Precision medicine; polymorphism; patient care planning; pharmacokinetics; personalised therapies

**Author for correspondence:**
Sarah N. Hilmer,
Email: sarah.hilmer@sydney.edu.au

# Polypharmacy and precision medicine

Kenji Fujita , Nashwa Masnoon , John Mach , Lisa Kouladjian O'Donnell and Sarah N. Hilmer

Departments of Clinical Pharmacology and Aged Care, Kolling Institute, Faculty of Medicine and Health, The University of Sydney and the Northern Sydney Local Health District, Sydney, NSW, Australia

## Abstract

Precision medicine is an approach to maximise the effectiveness of disease treatment and prevention and minimise harm from medications by considering relevant demographic, clinical, genomic and environmental factors in making treatment decisions. Precision medicine is complex, even for decisions about single drugs for single diseases, as it requires expert consideration of multiple measurable factors that affect pharmacokinetics and pharmacodynamics, and many patient-specific variables. Given the increasing number of patients with multiple conditions and medications, there is a need to apply lessons learned from precision medicine in monotherapy and single disease management to optimise polypharmacy. However, precision medicine for optimisation of polypharmacy is particularly challenging because of the vast number of interacting factors that influence drug use and response. In this narrative review, we aim to provide and apply the latest research findings to achieve precision medicine in the context of polypharmacy. Specifically, this review aims to (1) summarise challenges in achieving precision medicine specific to polypharmacy; (2) synthesise the current approaches to precision medicine in polypharmacy; (3) provide a summary of the literature in the field of prediction of unknown drug–drug interactions (DDI) and (4) propose a novel approach to provide precision medicine for patients with polypharmacy. For our proposed model to be implemented in routine clinical practice, a comprehensive intervention bundle needs to be integrated into the electronic medical record using bioinformatic approaches on a wide range of data to predict the effects of polypharmacy regimens on an individual. In addition, clinicians need to be trained to interpret the results of data from sources including pharmacogenomic testing, DDI prediction and physiological-pharmacokinetic-pharmacodynamic modelling to inform their medication reviews. Future studies are needed to evaluate the efficacy of this model and to test generalisability so that it can be implemented at scale, aiming to improve outcomes in people with polypharmacy.

## Impact statement

Achieving precision medicine in polypharmacy presents complex challenges due to the vast number of factors that influence drug use and response. Electronic Clinical Decision Support Systems have been developed to optimise polypharmacy and reduce medication-related harm in older adults. However, there is limited ability of these systems to account for complex, multidimensional interactions in a patient and to incorporate patient-specific goals of care. To address this, a novel approach to integrating precision medicine into medication reviews for patients with polypharmacy has been proposed. This approach applies bioinformatic techniques to a wide range of the patient's clinical, biological and drug data, to predict the effects of polypharmacy regimens on an individual. Implementing this model in routine clinical practice will require the integration of comprehensive intervention bundles into the electronic medical record, and training of healthcare professionals to interpret the results of data from sources, that include pharmacogenomic testing, drug–drug interaction prediction and physiological-pharmacokinetic-pharmacodynamic modelling, to inform their medication reviews. This proposed model could maximise the effectiveness of disease treatment and prevention while minimising harm from medications by systematically considering relevant demographic, clinical, genomic and environmental factors in making treatment decisions. Future research should aim to tailor these tools to specific patient populations, demonstrating long-term clinical outcomes relevant to patients' goals of care through informed shared decision-making

## Introduction

Precision medicine aims to maximise benefit and minimise harm from medicines, by considering relevant demographic, clinical, genomic and environmental factors in treatment decisions. While individualisation of treatment is a longstanding principle of good prescribing, precision medicine provides a framework to do so systematically (Peck, 2018; Rongen et al., 2021). Precision medicine is complex, even for decisions about single drugs for single diseases. It requires expert

consideration of multiple measurable factors that affect pharmacokinetics (PK) and pharmacodynamics (PD), and many patient-specific variables. Advances in therapeutic drug monitoring, pharmacogenomics, bioinformatics and physiological-pharmacokinetic-pharmacodynamic modelling have facilitated implementation of precision medicine over the past decade.

There are great opportunities to apply the lessons learned from precision medicine in single drug/disease management to optimisation of polypharmacy (most commonly defined as the use of five or more medicines) (Masnoon et al., 2017). Precision medicine for optimisation of polypharmacy is particularly challenging because of the need to consider an enormous number and complexity of interacting factors that influence drug use and response (Rongen et al., 2021). Polypharmacy usually occurs in older people with multimorbidity and a lifetime of accumulated environmental factors that influence response to medicines. These factors must be considered when prescribing, along with drug interactions, pharmacogenomics and the patient's therapeutic goals, which often extend beyond single disease prevention or management.

There is scope to bring together the technological framework of precision medicine as it has been applied to single drug/disease management, with emerging data on polypharmacy from bench, patient and population studies, and from development and use of clinical decision support systems for review of polypharmacy, to guide optimisation of polypharmacy in older people.

Given the complexity of the precision medicine process in polypharmacy, this narrative review aims to provide the latest research findings to achieve precision medicine in the context of polypharmacy. Specifically, this review aims to (1) summarise challenges in achieving precision medicine specific to polypharmacy; (2) synthesise the current approaches to precision medicine in polypharmacy; (3) provide a summary of the literature in the field of prediction of unknown drug–drug interaction (DDI) and (4) propose a novel approach to provide precision medicine for patients with polypharmacy.

## What makes precision medicine in polypharmacy challenging?

### Ageing

Older adults are at a high risk of suboptimal medication use, which includes overuse, underuse and misuse of medications, and this can lead to adverse drug events (ADE) and adverse health outcomes. The factors that lead to suboptimal medication use by older adults can be attributed to ageing-related factors, such as multimorbidity, frailty and changes in PK and PD. Prescribing medications must be a precise balance between minimising the number of medications and using all medications that will be beneficial at the optimal dose for the patient, whilst accounting for age-related factors (Steinman et al., 2006).

Multimorbidity, defined as the presence of multiple concurrent medical conditions, is more common with age and is associated with high mortality, reduced functional status and increased hospitalisation. In a cross-sectional study conducted in Scotland, 42.2% of all patients had one or more morbidities and 23.2% were multimorbid (Barnett et al., 2012). By 2035, approximately 17% of the UK population is projected to have four or more chronic conditions (Pearson-Stuttard et al., 2019). The challenge with multimorbidity and medication management comes with the application of clinical guidelines; most clinical guidelines are built on evidence-based medicine and are designed for the treatment of single diseases,

and often overlook medication management in multimorbid older adults. Application of single-disease clinical guidelines in multimorbid older adults can lead to overtreatment. A Norwegian qualitative study with general practitioners (GPs) that explored experiences with and reflections upon the consequences of applying multiple clinical guidelines in older multimorbid adults, found that when GPs focus on person-centred care and refrain from complying with clinical guidelines the risks associated with polypharmacy and overtreatment can be reduced (Austad et al., 2016).

Like multimorbidity, frailty adds more complexity to precisely balancing medication management for older adults. Frailty is a complex geriatric syndrome and a state of vulnerability, which can result in decreased physiological reserve (Clegg et al., 2013). Frailty is common in later life, with prevalence between 10 and 14% for community-dwelling older adults, and up to 50% for older people living in residential aged care facilities (Collard et al., 2012; Kojima, 2015). A systematic review that analysed the evidence and interplay between polypharmacy and frailty in older adults, identified that the association between frailty and polypharmacy may be complex and bidirectional, but polypharmacy is recognised as a major contributor to the development of frailty (Gutierrez-Valencia et al., 2018). Many tools have been developed to help clinicians identify inappropriately prescribed medications in older people with polypharmacy (e.g., STOPPFrail) (Thompson et al., 2019). However, limited clinical studies demonstrate improvements in frailty when deprescribing medications (Ibrahim et al., 2021). A study conducted in aged mice with chronic polypharmacy found that deprescribing high-risk medications attenuated frailty, identifying that there is potential to reduce the effects of frailty by appropriately and precisely managing polypharmacy (more details provided in section 'Limitations of evidence from human studies and role of preclinical models') (Mach et al., 2021a).

The physiological changes that occur in ageing can affect PK and PD, which in turn can impose considerable variability in medication management for older adults with polypharmacy. Age-related changes in PK/PD include changes in drug absorption from the gastrointestinal tract, plasma protein binding, drug distribution, reduced hepatic metabolism and clearance, altered renal function, changes to receptors and voltage-gated channels, and changes to the autonomic nervous system (Hilmer et al., 2007b). These changes may be exaggerated in frail older people, although the data is very limited (Hilmer and Kirkpatrick, 2021). Changes in PK/PD also may make older adults with polypharmacy more vulnerable to ADEs. Given the wide variability in response to medicines by older adults, with the added complexity of under-representation of older adults in PK and PD studies and limited clinical trial data, there has been recent debate about the role of PK and PD studies for frail older adults to inform medication management (McLachlan et al., 2009; Mangoni et al., 2013; Liau et al., 2021). There is also scope for PK/PD monitoring, including therapeutic drug monitoring, in clinical practice to improve precision medicine in highly variable older people with polypharmacy.

### Interactions

Precision medicine in polypharmacy needs to consider the increased complexity of drug interactions when multiple drugs are used concurrently, particularly in people with multimorbidity. These include PK and PD DDIs, drug–gene interactions, drug–disease interactions, drug–food interactions, drug–lifestyle interactions and drug–microbiome interactions (Johnell and Kiarin,

2007; Guthrie et al., 2015). The challenges of each of these factors as they relate to polypharmacy are described below. Furthermore, these interactions influence each other, which will require advanced bioinformatics to fully understand and predict the overall effect on an individual patient. This concept is illustrated in the case described in Box 1.

Polypharmacy is the greatest risk factor for DDIs, which may be PK, PD or both. DDIs can occur between prescribed drugs, over the

---

– Educate patient on spending time in sun with skin exposed to increase vitamin D and increase dose of cholecalciferol.
– Refer to dietician to address malnutrition and calcium intake.
– Refer to physiotherapist for exercise program to reduce risk of falls.
– Screen for/manage other falls risk factors (e.g., vision, footwear, environment) and refer for further assessment and management as required.

---

**Box 1.** Case study of patient with polypharmacy demonstrating complexity of multiple types of drug interactions.

Mrs. C.P. is an 88-year-old woman living independently at home. She is malnourished and has had recurrent falls. Diagnoses include osteoarthritis, osteoporosis, gastro-oesophageal reflux disease, hypertension, ischaemic heart disease and depression. Her medications are paracetamol 1 g tds, alendronate 70 mg weekly, cholecalciferol 1,000 units daily, omeprazole 40 mg daily, lisinopril 10 mg daily, metoprolol 25 mg bd, aspirin 100 mg daily and citalopram 20 mg daily. She has no known drug allergies.
Interactions include:

– Drug–drug interactions
  ○ Pharmacokinetic: citalopram increases concentration of metoprolol by inhibiting CYP2D6; omeprazole increases concentration of citalopram by inhibiting CYP2C19.
  ○ Pharmacodynamic: citalopram and aspirin have additive effects reducing haemostasis; alendronate and aspirin have additive effects damaging gastrointestinal mucosa; aspirin may reduce the effects of lisinopril in a dose-dependent manner (usually at aspirin dose >100 mg daily).
– Drug–gene interactions
  ○ CYP2D6 polymorphisms affect the clearance of metoprolol, with implications for the effect size of the interaction between metoprolol and citalopram. Similarly, CYP2C19 polymorphisms affect the clearance of citalopram, with implications for the impact of the interaction between citalopram and omeprazole.
– Drug–disease interactions
  ○ Alendronate oesophageal toxicity more severe with gastro-oesophageal reflux disease.
  ○ Citalopram more likely to cause long QT syndrome with background of cardiac disease.
– Drug–food interactions
  ○ Malnutrition is a risk factor for paracetamol hepatotoxicity.
  ○ Inadequate calcium in diet for alendronate to be efficacious.
– Drug–lifestyle interactions
  ○ Minimal sunlight exposure due to fear of falling, resulting in low vitamin D at current dose of cholecalciferol, and consequently poor calcium absorption, reducing efficacy of alendronate.
– Drug–microbiome interactions
  ○ Drugs, diagnoses, diet, age and lifestyle factors all associated with changes in microbiome, which can alter pharmacokinetics and pharmacodynamics of other drugs.
– Drug–geriatric syndrome interactions
  ○ Lisinopril, metoprolol and citalopram may all increase risk of falls; at increased risk of fall-related injury with underlying osteoporosis.

Clinical Recommendations:

– Cease alendronate due to gastro-oesophageal reflux and malnutrition. As ongoing high risk of fracture, change to zoledronic acid or denosumab. Approximately 4 weeks after cease alendronate, trial deprescribing omeprazole.
– Check heart rate, blood pressure and postural blood pressure. If bradycardia and/or hypotension/postural hypotension, then reduce dose metoprolol, noting that clearance may be reduced by interaction with citalopram, especially if CYP2D6 poor metaboliser. If patient does not have bradycardia but does have hypotension/postural hypotension, then reduce dose of lisinopril.
– Review indication for citalopram and check for toxicity (ECG for QT interval, postural hypotension and serum sodium). If citalopram is currently indicated, then reduce dose if evidence of toxicity; if no longer required then deprescribe.

---

counter drugs, complementary and alternative medicines; and affect drugs used acutely, chronically and intermittently. Any change in drug or dose (including prescribing or deprescribing) can impact existing DDIs. As described in section 'Use of machine learning for predicting polypharmacy interactions and effects', drug interactions are generally only assessed for drug pairs, and the effects of interactions beyond drug pairs remain poorly understood. Recent attempts have been made to understand the impact of PD DDIs involving multiple drugs in the setting of polypharmacy, for example through tools to measure anticholinergic burden (Salahudeen et al., 2015).

Drug–gene interactions are common and there is increasing understanding of the role of pharmacogenomics in determining PK and PD variability. While this is only one of many factors that influence variability in drug response in older adults with polypharmacy, it remains important (Dücker and Brockmöller, 2019). For example, drug clearance through a particular pathway may be affected by a genetic polymorphism, as well as by other drugs that inhibit or induce the pathway. This is a complex two-way relationship, whereby both factors may work in the same or in opposite directions. Furthermore, in a patient with polypharmacy, there may be multiple polymorphisms affecting multiple pathways, and multiple drugs each using these pathways for clearance. A recent review of the impact of pharmacogenomic testing for PK factors in patients with polypharmacy identified six studies of variable quality, and five reported improved clinical outcomes or reduced drug/health utilisation outcomes (Meaddough et al., 2021).

Drug–disease interactions are extremely common in people with polypharmacy, since it often goes hand in hand with multimorbidity. The concept of 'therapeutic competition' addresses the clinical challenge of selecting which condition to treat, when treatment of one of a patient's conditions worsens another of their conditions. It is estimated that one in five older Americans receive medications that may adversely affect co-existing conditions (Lorgunpai et al., 2014).

Drug–food interactions include a wide range of PK and PD interactions between specific drugs and foods, the effects of overall nutritional state on drug PK and PD, and the effects of drugs on nutrition (Schmidt and Dalhoff, 2002). Medications can either stimulate appetite, resulting in obesity, or more commonly in people with polypharmacy, can cause nausea and reduce appetite resulting in malnutrition (Fávaro-Moreira et al., 2016). Recent studies have used data mining to predict and evaluate food–drug interactions (Rahman et al., 2022). This method is highly applicable to people with polypharmacy.

Drug–lifestyle interactions cover interactions with diverse factors such as alcohol, smoking and exercise. There are well-characterised PK and PD drug interactions with alcohol and smoking, including the impact of therapeutic drugs on the clearance of alcohol and nicotine, and impact of alcohol and nicotine on drug clearance. Therapeutic drugs can increase or decrease exercise

capacity through cardiorespiratory effects or neuromuscular effects. Anabolic exercise can be used to counter sarcopaenia induced by drugs such as prednisone, or strength and balance training can be used to reduce susceptibility to falls risk-increasing drugs (The Agency for Clinical Innovation, 2021).

The bidirectional interactions between a wide range of drugs and microbiome have recently been characterised and provide some explanation for previously unexplained inter-individual variability in drug response (Weersma et al., 2020). A wide range of therapeutic drugs affects the microbiome in different ways, as do age, sex, disease, frailty, dementia and polypharmacy. Recently the microbiome signature of polypharmacy was characterised in observational studies in older people (Nagata et al., 2022). Interventional studies in mice found changes in microbiome with the single polypharmacy regimen tested, which was partially reversed after deprescribing (withdrawal) in old age (Gemikonakli et al., 2022). More research is needed to understand the additive or synergistic effects of the multiple medications in polypharmacy, along with effects of multimorbidity and other variables on the microbiome.

The interactions between drugs and geriatric syndromes, such as falls, frailty and confusion, are well-recognised in geriatric medicine. Drugs are considered the most reversible causes of these presentations (Avorn and Shrank, 2008). Polypharmacy itself is one of the strongest risk factors, with different drug classes more strongly associated with specific geriatric outcomes, often with evidence of a dose response and/or cumulative effects (Hilmer and Gnjidic, 2009).

### *Limitations of evidence from human studies and role of preclinical models*

Precision medicine requires consideration of a multitude of factors to tailor medication to an individual. The added complexity of considering the enormous number of combinations of drugs in different polypharmacy regimens, along with different combinations of factors within the individual can be overwhelming in terms of interventional clinical trial design. Application of bioinformatics to this challenge is an emerging strategy, described in section 'Use of machine learning for predicting polypharmacy interactions and effects'.

Randomised trials have investigated the effects of polypharmacy for single diseases. For example, the use of multiple drugs is endorsed in guidelines for conditions such as tuberculosis, HIV, heart failure, ischaemic heart disease and diabetes. Many of the large randomised controlled studies that inform these guidelines include subgroup analyses by age, sex, comorbidities and more recently by frailty (Dewan et al., 2020; Nguyen et al., 2021). Another source of evidence has been subgroup analyses of clinical trials investigating treatment of a single disease according to baseline polypharmacy in the participants (Jaspers Focks et al., 2016). This gives information on the effects of polypharmacy on the efficacy and safety of monotherapy for a single disease, but such analyses have not extended to consider the impact of other factors that inform personalised medicine.

Factors that influence the effects of polypharmacy can be evaluated indirectly through observational studies in populations of older adults. The ability of this data to inform precision medicine is currently very limited, with a recent review highlighting the lack of data even on the effects of sex and gender on polypharmacy outcomes, let alone the myriad of other individual factors (Rochon et al., 2021).

Interventional trials of polypharmacy that consider different baseline characteristics in subgroups would give data comparable to the data that informs precision medicine for monotherapies. It is not ethical or feasible to conduct interventional randomised controlled trials to evaluate the effects of polypharmacy used for multimorbidity in older adults. Recently, a polypharmacy mouse model was developed, to understand the effects of polypharmacy on key outcomes in old age, and to investigate the effects of common factors that might influence these effects, such as the composition of the polypharmacy regimen, age and sex. An assay to measure pharmacokinetic variability as a factor in personalised medicine was also developed (Mach et al., 2021b). Application of this model by our laboratory and international collaborators, has shown that polypharmacy causes frailty and functional impairment, with greater effects seen with drug regimens with higher anticholinergic and sedative load (measured using Drug Burden Index, DBI), greater impairment in old age, and different patterns of physical and cognitive impairments between males and females. These findings are shown in Table 1. There are now opportunities to analyse proteomics, transcriptomics, metabolomics and microbiome data from these well-characterised phenotypes, using systems biology, to identify biomarkers that predict PK and PD responses to polypharmacy and deprescribing. These could be used to inform precision medicine for people with polypharmacy, for example, by integration into physiological-based PK-PD modelling.

## Current approach to precision medicine in polypharmacy

### *Use of decision support tools*

#### *Criteria included in existing decision support tools and limitations*
Several decision support tools and guidelines have been developed to optimise polypharmacy and reduce medication-related harm in older adults. Some tools simply provide a list of potentially inappropriate medications (PIM) in the older population such as the PRISCUS list (Latin for 'old and vulnerable') (Holt et al., 2010). In contrast, other tools have additional criteria for identifying PIMs, such as interaction between drugs and diseases for example the Beers criteria (Fick et al., 2019) and Screening Tool of Older Persons' Prescriptions (STOPP) and Screening Tool to Alert to Right Treatment (START) (O'Mahony et al., 2015). It is important to consider patient-specific factors such as dose appropriateness for the particular patient. However, a systematic review of different polypharmacy tools (Masnoon et al., 2018) found that whilst 64.3% of tools mention dosing, only 2.4% consider specific doses being used, such as the DBI (Hilmer et al., 2007a). Development of electronic Clinical Decision Support Systems (CDSS) has been identified as a key facilitator in uptake of these tools in busy clinical practice and a step towards precision medicine in polypharmacy.

Table 2 summarises different electronic CDSS published in the last 10 years, based on existing polypharmacy tools. Studies were identified (Mouazer et al., 2022) and data were extracted (Masnoon et al., 2018) using the search strategy utilised in previous literature, with the date range set to the last 10 years (January 2012 to October 2022).

In terms of criteria considered by different electronic CDSSs to guide polypharmacy review, all tools require a patient's medication list (Holt et al., 2010). Some tools apply other additional criteria to tailor the output to the specific patient, such as health conditions or disorders, laboratory test results and pharmacogenomic data (Mouazer et al., 2022). For example, the Software ENgine for the

**Table 1.** The effects of polypharmacy on global health outcomes in a mouse model: impact of drug regimen, age and sex

| Study population and intervention | Outcomes | Application to precision medicine | References |
| --- | --- | --- | --- |
| Young and old male mice, 4–6 weeks of polypharmacy* versus control | Change in physical function tests | Age effects | Huizer-Pajkos et al., 2016 |
| Middle-aged male mice, 12 months of one of three polypharmacy regimens (five drugs) with different Drug Burden Index (DBI)**, monotherapies or control | Change in physical function tests and frailty: functional impairment/frailty related to DBI, not simply polypharmacy Effects reversible with deprescribing | Drug regimen effects | Mach et al., 2021a |
| Young adult male mice, 8 weeks polypharmacy^ or control | Change in exploration and spatial working memory | Sex effects on cognition | Eroli et al., 2020 |
| Young adult female mice, 8 weeks polypharmacy^ or control | Change in object recognition and fear-associated contextual memory No effects on exploration and spatial working memory | Sex effects on cognition | Francesca et al., 2021 |
| Young and old male and female mice, 6 weeks high DBI polypharmacy** or control | Changes in physical function tests Serum drug/metabolite concentrations | Age and sex effects on physical function Consideration of pharmacokinetic factors | Wu et al., 2021 |
| Young and old male and female mice, 6 weeks high DBI polypharmacy** or control | Changes in behaviour over 23 hours | Age and sex effects on diurnal patterns in behaviour | Tran et al., 2022 |
| Male mice aged 24 months, 3 polypharmacy regimens** or control | No effects of polypharmacy regimens on serum inflammatory markers in mice | Inflammatory biomarkers not independently affected by polypharmacy | Wu et al., 2022 |
| Male mice aged 12–24 months, high DBI polypharmacy**, high DBI polypharmacy deprescribed or control | Polypharmacy alters gut microbiome differently to age effects. Partially reversible with deprescribing | Age, polypharmacy and deprescribing effects on microbiome may affect pharmaco-microbiomics | Gemikonakli et al., 2022 |

*Note:* Drugs administered in polypharmacy regimens were *simvastatin, metoprolol, omeprazole, paracetamol, citalopram; **zero DBI polypharmacy: simvastatin, metoprolol, omeprazole, paracetamol, irbesartan; low DBI polypharmacy: simvastatin, metoprolol, omeprazole, paracetamol, citalopram; high DBI polypharmacy: simvastatin, metoprolol, oxybutynin, oxycodone, citalopram; each drug from high DBI regimen also administered as monotherapy; ^ simvastatin, metoprolol, aspirin, paracetamol, citalopram.

Assessment and optimisation of drug and non-drug Therapy in Older peRsons (SENATOR) uses STOPP START (O'Mahony et al., 2015), the MedSafer system uses Beers criteria (Fick et al., 2015), STOPP(O'Mahony et al., 2015) and evidence-based recommendations from Choosing Wisely Canada (McDonald et al., 2019; Baysari et al., 2021), and the Goal-directed Medication review Electronic Decision Support System (G-MEDSS) uses The DBI Calculator (Kouladjian et al., 2016; Kouladjian O'Donnell et al., 2022).

There are limitations in terms of criteria considered by existing tools. Firstly, real-world patients are complex, with multiple conditions and medications. Current tools however lack intelligent algorithms to account for multiple complex interactions in a patient, for example, different drug–food and drug–gene interactions (Finkelstein et al., 2016; Mehta et al., 2021; Westerbeek et al., 2021; Damoiseaux-Volman et al., 2022; Mouazer et al., 2022). Additionally, an important consideration in precision medicine is distinguishing between theoretical and clinically relevant drug interactions for a particular patient, which is another limitation. Caring for real-world patients often requires managing conflicting recommendations from different guidelines in the same patient but existing tools lack algorithms to provide tailored decision support in these scenarios. Whilst there is data suggesting that machine learning, which refers to the use of algorithms and statistical models to analyse and interpret medical data in order to generate predictions or insights that can inform clinical decision-making, may be a promising approach to developing polypharmacy CDSS (Corny et al., 2020), previous research has stated that most current

electronic CDSS have not used machine learning algorithms to target output signals (Mouazer et al., 2022). Lastly, an important aspect of precision medicine is making therapeutic decisions whilst considering the patient's specific goals of care. However, goals of care are not routinely considered by different polypharmacy CDSS (Finkelstein et al., 2016; Mehta et al., 2021; Mouazer et al., 2022). Recently, some CDSS have integrated goals of care assessment with other tools to address polypharmacy (Mangin et al., 2021; Kouladjian O'Donnell et al., 2022).

### Outcome evaluation of existing decision support tools and limitations

Previous research has identified three key outcome measures when evaluating polypharmacy optimisation CDSS: (1) impact on clinical outcomes and impact on clinical practice, (2) efficiency in terms of time spent and (3) user satisfaction (Mouazer et al., 2022). There is significant heterogeneity in the study design, methods and outcome measures for studies evaluating different polypharmacy CDSS (Mouazer et al., 2022). Few studies have used randomised controlled trials. Most studies have found effectiveness based on impact on clinical practice, namely changes in prescribing. For example, using the MedSafer system during acute hospitalisation was found to increase deprescribing at discharge but no significant impact was found on adverse drugs events within 30 days of discharge (McDonald et al., 2019, 2022). It is important to demonstrate impact on long-term clinical outcomes, which are relevant to patients as per their goals of care, with specific focus on shared decision making.

Precision medicine relies on making therapeutic decisions tailored to the specific patient. However, more research is needed to determine how different polypharmacy CDSS can be tailored to specific populations including different types of medicines, chronic conditions, age groups, ethnic backgrounds, prognosis, laboratory results and pharmacogenomics.

### Use of machine learning for predicting polypharmacy interactions and effects

Understanding polypharmacy effects is an essential step to optimise medication regimens. However, most of the known polypharmacy effects are highly variable and non-specific and usually not detectable in clinical trials (Bansal et al., 2014). Given the vast number of drug combinations, neither experiments nor clinical trials can investigate the effects of DDIs for all possible combinations due to time and cost. Therefore, computational methods have been developed for predicting unknown DDIs that cause effects that cannot be attributed to single drugs alone (Han et al., 2021).

Generally, the methods for predicting DDIs are divided into two categories: (1) prediction of DDIs, and (2) prediction of specific types of DDIs (Han et al., 2021). The first category predicts whether a pair of drugs will cause DDI. This category can be further classified into similarity-based and classification-based methods. The idea behind the similarity-based method is that if drug A and drug B cause a DDI, drug C similar to drug A should also interact with drug B. Different types of Drug–drug similarities are used based on molecular structures, side effects, pharmacology and biological elements (e.g., carriers, transporters, enzymes and targets) (Gottlieb et al., 2012; Vilar et al., 2012; Ferdousi et al., 2017). In contrast, classification-based methods treat the prediction of DDI between paired drugs as a binary classification task. Known drug pairs with DDI and drug pairs with non-DDI are used as positive and negative cases, respectively, to build classification models such as logistic regression, naïve Bayes, $k$-Nearest neighbours, and support vector machine (Cheng and Zhao, 2014; Huang et al., 2014; Li et al., 2015; Kastrin et al., 2018). The second category predicts whether specific DDI will be caused by a pair of drugs. Zitnik et al. (2018) proposed a graph convolutional neural network for multi-relational link prediction called Decagon, which is one of the most well-established models. This multimodal graph includes 645 drugs and 19,085 proteins as nodes, and 4,651,131 DDIs, 715,612 protein–protein interactions, and 18,596 drug–protein interactions as edges. The model predicts associations between pairs of drugs and the specific side effects in the pair as a link prediction task. Since the Decagon model was proposed, other models have been developed for specific DDIs prediction (Nováček and Mohamed, 2020; Bang et al., 2021; Masumshah et al., 2021).

To achieve better prediction accuracy of the models, various information needs to be extracted from multiple sources. Previous studies have reported that information on the presence and severity of DDIs often vary among databases, which affect the result of model performance (Abarca et al., 2004; Wang et al., 2010; Saverno et al., 2011). For example, the total number of reported DDIs for 12 commonly prescribed drugs was 1,226 in Kyoto Encyclopedia of Genes and Genomes and 1,533 in DrugBank (Ferdousi et al., 2017). Considering that the number of reported DDIs in these databases increases with each update even between previously existing drug pairs, it is not possible to tell whether drug pairs not reported as having DDIs are true negatives or not-yet-known positives (i.e., false negatives). Therefore, special attention needs to be paid to which data sources were used to build and compare prediction models.

There are a few limitations in the current DDI prediction models. First, the current DDI prediction models only consider effects for two drugs. There is little knowledge of interaction effects unique to three or more drugs that do not occur with two or fewer drugs. Given that most patients with multimorbidity are prescribed more than two drugs, DDI prediction for more than two drugs is important. Secondly, the current DDI prediction models do not consider individual-level predictors (e.g., demographic, clinical and genetic information), as well as detailed drug regimens (e.g., administration route and dosage). Considering that the management of complex drug interactions (e.g., DDIs, drug–gene interactions) as combined parameters affecting drug response is a complex task particularly in older patients with polypharmacy, a comprehensive medication review process will be necessary using a multifaceted intervention bundle with accompanying stewardship program.

### Future direction

In recent years, advances in technology and data analysis for detailed clinical, biological and molecular phenotyping have helped build the evidence for the efficacy of pharmacogenomics to guide prescribing for therapies such as anticoagulants (Roberts et al., 2012; Pirmohamed et al., 2013), antidepressants (Greden et al., 2019; Ruaño et al., 2020), antipsychotics (Herbild et al., 2013) and statins (Peyser et al., 2018). Based on such evidence, the Clinical Pharmacogenetics Implementation Consortium (CPIC) has published guidelines on how to adjust drugs based on genetic test results (Relling and Klein, 2011). However, these guidelines have been predominantly guided by studies that focused on single drug–gene or disease–gene pairs (O'Shea et al., 2022). Given the complexity of pharmacogenomic interactions amongst multiple drugs and proteins, the effectiveness of pharmacogenetic interventions in adults with polypharmacy needs to be established.

To maximise the potential of pharmacogenomics in routine care of adults with polypharmacy, several elements that have been reported as key facilitators could be applied, including (1) a proper infrastructure to integrate pharmacogenomics into the workflow of physicians and pharmacists (van der Wouden et al., 2017; Slob et al., 2018), (2) improvement in physicians', pharmacists' and patients' awareness and education about pharmacogenomics (Jansen et al., 2017; Tonk et al., 2017) and (3) clear clinical pathways and allocation of responsibilities between healthcare providers about who should interpret pharmacogenomics results and communication with patients (Finkelstein et al., 2016; Lanting et al., 2020).

In addition, as previously discussed, polypharmacy is highly prevalent in older adults who are likely to experience adverse events due to factors other than genetic polymorphisms, including multimorbidity, frailty and lifestyle (McLachlan et al., 2009). The effects of these factors can be considered through analysis of a wide range of big data, ranging from the clinical, functional and socio-demographic data captured in health records, to the variability of the microbiome. Therefore, multiple factors need to be evaluated to optimise polypharmacy drug regimens. While existing CDSS for polypharmacy consider some drug or patient factors, as outlined in Table 2, there is potential to integrate these with factors identified from pre-clinical studies, pharmacogenomics, drug interaction data and clinical data including therapeutic drug monitoring, to guide precision medicine for patients with polypharmacy.

**Table 2.** Electronic clinical decision support systems to optimise polypharmacy

| System, References | Country and year | User | Knowledge base | Input data | DDIs | DDSIs | DGIs | Dosing | Impact of renal function on drug clearance |
|---|---|---|---|---|---|---|---|---|---|
| CheckUP, Linkens et al., 2022 | Netherlands 2022 | Physicians Pharmacists | STOPP START (O'Mahony et al., 2015) | – Drugs<br>– Age<br>– Gender<br>– Laboratory test results | Y | Y | N | Y[a] | Y |
| Frutos et al., 2022 | Argentina 2022 | General practitioners | – Beers (Fick et al., 2019) | – Drugs<br>– Age<br>– Gender | Y | Y | N | Y[a] | Y |
| MediQuit, Junius-Walker et al., 2021 | Germany 2022 | Physicians | – Systematic review on deprescribing guides in primary care | – Drugs<br>– Medical conditions<br>– Frailty | NS | NS | NS | NS | NS |
| Persell et al., 2022 | USA 2022 | Physicians | – Beers (Fick et al., 2015)<br>– STOPP (O'Mahony et al., 2015)<br>– National Action Plan for Adverse Drug Event Detection (U.S. Department of Health and Human Services, 2014) | – Drugs<br>– Medical conditions | Y | Y | N | Y[a] | Y |
| Singhal et al., 2022 | USA 2022 | Clinicians | – Beers (Fick et al., 2015) | – Drugs | N | N | N | N | N |
| Bittmann et al., 2021 | Germany 2021 | Prescribers | – AiDKlinik (Dosing GmbH, 2022) | – Drugs | Y | N | N | N | N |
| DBI Hospital Intervention Bundle, Baysari et al., 2021; Masnoon et al., 2022 | Australia 2021 | Healthcare professionals | – DBI (Hilmer et al., 2007a)<br>– Clinician deprescribing guides (NSW Therapeutic Advisory Group, 2021)<br>– Consumer information leaflets (NSW Therapeutic Advisory Group, 2021)<br>– Education module on deprescribing (Health Education and Training, 2018) | – Drugs | N | N | N | Y[b] | N |
| OPERAM, Blum et al., 2021 | Europe 2020 | Physicians Pharmacists | – STOPP START (Gallagher et al., 2011) | – Drugs<br>– Medical conditions<br>– Laboratory test results | Y | Y | N | Y[a] | Y |
| Rogero-Blanco et al., 2020 | Spain 2020 | Physicians | – Beers (Fick et al., 2015)<br>– STOPP START (Delgado Silveira et al., 2015) | – Drugs<br>– Medical conditions | Y | Y | N | Y[a] | Y |
| G-MEDSS, G-MEDSS, 2019; Kouladjian O'Donnell et al., 2022 | Australia 2019 | Healthcare professionals | – DBI (Hilmer et al., 2007a)<br>– Goals of care<br>– rPATD (Reeve et al., 2016) | – Drugs | N | N | N | Y[b] | N |
| Zwietering et al., 2019 | Netherlands 2019 | Clinicians | – STOPP START (O'Mahony et al., 2015) | – Drugs<br>– Laboratory test results | Y | Y | N | Y[a] | Y |
| García-Caballero et al., 2018 | Spain 2018 | Physicians | – STOPP (O'Mahony et al., 2015) | – Drugs | Y | Y | N | Y[a] | Y |
| Kim et al., 2018 | USA 2018 | Pharmacists | – Not specified | – Drugs<br>– Pharmaco-genetic test results | Y | Y | Y | Y[a] | N |
| Liu et al., 2018 | USA 2018 | NS | – UpToDate (Wolters Kluwer, 2022)<br>– Indiana University portal (Flockhart, 2021) | – Drugs<br>– Genetic data | Y | N | Y | N | N |

| System, References | Country and year | User | Knowledge base | Input data | DDIs | DDSIs | DGIs | Dosing | Impact of renal function on drug clearance |
|---|---|---|---|---|---|---|---|---|---|
| | | | – SuperCyp (Preissner et al., 2010)<br>– PharmGKB (Whirl-Carrillo et al., 2012)<br>– SNPedia (Cariaso and Lennon, 2012) | | | | | | |
| Johansson-Pajala et al., 2018 | Sweden 2017 | Physicians Registered nurses | – Beers (Fick et al., 2015)<br>– STOPP START (O'Mahony et al., 2015) | – Drugs<br>– Medical conditions | Y | N | N | Y[a] | Y |
| MedSafer, McDonald et al., 2019 | Canada 2017 | Clinicians | – Beers (Fick et al., 2015)<br>– STOPP (O'Mahony et al., 2015)<br>– Choosing Wisely Canada (The American Board of Internal Medicine Foundation, 2022)<br>– Literature review on deprescribing | – Drugs<br>– Medical conditions<br>– Frailty | Y | Y | N | Y[a] | Y |
| PIM-Check, Blanc et al., 2018 | Switzerland 2017 | Junior hospital physicians and pharmacists | – Internal PIMs list | – Drugs<br>– Medical conditions | Y | Y | N | Y[a] | Y |
| Verdoorn et al., 2018 | Netherlands 2017 | Pharmacists | – Beers (Fick et al., 2015)<br>– STOPP START (O'Mahony et al., 2015) | – Drugs | Y | Y | N | Y[a] | Y |
| PRIMA-eDS, Sönnichsen et al., 2016 | Finland 2016 | Physicians | – EU(7) PIM list (Renom-Guiteras et al., 2015)<br>– SFINX (Böttiger et al., 2009)<br>– RISKBASE (Medbase, 2015b)<br>– RENBASE (Medbase, 2015a) | – Drugs<br>– Medical conditions<br>– Symptoms<br>– Biometric measurements (such as body mass index and blood pressure)<br>– Laboratory test results | Y | N | N | Y[a] | Y |
| SENATOR, Dalton et al., 2020 | Europe 2016 | Clinicians | – STOPP START (O'Mahony et al., 2015)<br>– British National Formulary<br>– SafeScript<br>– CIRS-G (Miller et al., 1992)<br>– ONTOP (Abraha et al., 2015) | – Drugs<br>– Medical conditions | Y | Y | N | Y[a] | Y |
| SMART, Alagiakrishnan et al., 2016 | Canada 2016 | Physicians Geriatricians | – Beers (Fick et al., 2015) | – Drugs | Y | Y | N | Y[a] | Y |
| TRIM, Fried et al., 2017 | USA 2016 | Pharmacists | – Beers (Fick et al., 2015)<br>– STOPP (O'Mahony et al., 2015)<br>– Medication Regimen Feasibility (Morisky et al., 2008)<br>– Renal dosing guidelines | – Age<br>– Drugs<br>– Medical conditions | Y | Y | N | Y[a] | Y |
| O'Sullivan et al., 2016 | Ireland 2015 | Pharmacists | – Beers (Fick et al., 2015)<br>– STOPP START (O'Mahony et al., 2015)<br>– PRISCUS list (Holt et al., 2010)<br>– Product information | – Drugs<br>– Medical conditions | Y | N | N | Y[a] | Y |
| GraphSAW, Holt et al., 2010 | Germany 2015 | Health professionals Researchers | – DrugBank (Knox et al., 2010)<br>– ABDA (Avoxa, 2009)<br>– KEGG (Kanehisa et al., 2012)<br>– SIDER (Kuhn et al., 2010) | – Drugs<br>– Medical conditions | Y | Y | N | N | N |

(*Continued*)

**Table 2.** (*Continued*)

| System, References | Country and year | User | Knowledge base | Input data | DDIs | DDSIs | DGIs | Dosing | Impact of renal function on drug clearance |
|---|---|---|---|---|---|---|---|---|---|
| STRIP Assistant, Meulendijk et al., 2015 | Switzerland and Netherlands 2015 | Physicians Pharmacists | – STOPP START (Meulendijk et al., 2015)<br>– Drug interaction guidelines | – Drugs<br>– Medical conditions<br>– Laboratory test results | Y | Y | N | Y[a] | Y |
| INTERcheck, Ghibelli et al., 2013 | Italy 2013 | Clinicians | – Beers (American Geriatrics Society 2012 Beers Criteria Update Expert Panel, 2012)<br>– ACB scale (Boustani et al., 2008)<br>– Drug–interaction database | – Drugs | Y | Y | N | Y[a] | Y |
| Grando et al., 2012 | USA 2012 | Not specified | – MRCI (George et al., 2004)<br>– Clinical guidelines for different diseases management | – Drugs<br>– Medical conditions | Y | N | N | Y[a] | Y |

*Note:* Studies were identified using the search strategy outlined by Mouazer et al. (2022), with the date range was set to the last 10 years (2012 to October 2022). Data items included in the table were guided by Masnoon et al. (2018) and Mouazer et al. (2022).
Y, Yes (characteristic considered by the CDSS); N, No (characteristic not considered by the CDSS), Y[a], mentions dosing only; Y[b], based predominantly on actual doses being used.
Abbreviation: ACB Scale, Anticholinergic Cognitive Burden Scale; DBI; Drug Burden Index; DDI; Drug–drug interaction; DDSI, Drug–disease interaction; DGI, Drug–gene interaction; MRCI, Medication Regimen Complexity Index; NS, not specified (unclear if characteristic considered by the CDSS); OPERAM, Optimising Therapy to Prevent Avoidable Hospital Admissions in Multimorbid Older Adults; PIM, Potentially Inappropriate Medication; PRISCUS, Latin for 'old and vulnerable'; rPATD, Revised Patients' Attitudes Towards Deprescribing; START, Screening Tool to Alert to Right Treatment; STOPP, Screening Tool of Older Persons' Prescriptions.

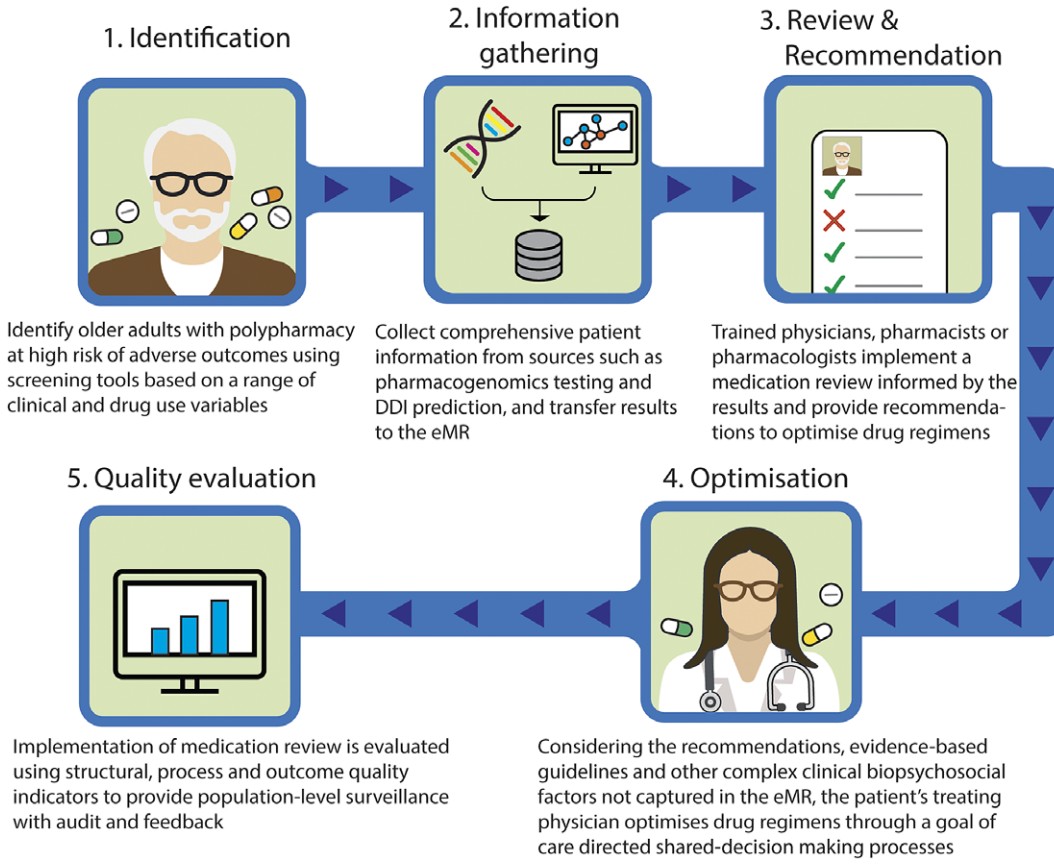

**Figure 1.** Novel approach to involve precision medicine for patients with polypharmacy. DDI, drug–drug interaction; eMR, electronic medical record.

Taking these into account, we outline a proposed approach to provide precision medicine as part of medication review for patients with polypharmacy (Figure 1). This model involves the following steps: (1) identify older adults with polypharmacy at high risk of adverse outcomes using screening tools based on a range of clinical and drug use variables; (2) collect comprehensive patient information from sources such as pharmacogenomics testing and DDI prediction, and transfer results to the eMR, allowing for computational analysis to predict outcomes; (3) trained physicians, pharmacists or pharmacologists implement a medication review for the patients informed by the results of pharmacogenomic testing, DDIs prediction, physiological-pharmacokinetic-pharmacodynamic modelling and routinely used care assessment data stored in the eMR (e.g., patient's medical conditions, hepatic/renal function, frailty, medications, any drug allergies or intolerances, results of any therapeutic drug monitoring). The review provides recommendations to optimise drug regimens (i.e., dose change, cease, start new therapies); (4) considering the recommendations, evidence-based guidelines and other complex clinical biopsychosocial factors not captured in the eMR, the patient's treating physician optimises drug regimens through a person-centred shared-decision making process. To facilitate deprescribing of inappropriate polypharmacy for older people, the use of comprehensive intervention bundles, such as training modules for healthcare providers, patient education leaflets and individualised goal attainment outcomes, may be effective (McDonald et al., 2022) and (5) implementation of medication review is evaluated using structural, process and outcome quality indicators to provide population-level surveillance with audit and feedback.

The novelty of this model lies in the implementation of medication review by a multidisciplinary team, based on the integrated results from a range of sources and their pharmacological expertise. Even if DDI prediction models demonstrate good predictive performance, the rationale behind their decisions is difficult to interpret, and expert interpretation is needed (Topol, 2019). The same applies to pharmacogenomic, therapeutic drug monitoring or pharmacological modelling data. Integrating data from these different sources including DDIs and pharmacogenomic factors requires complex clinical interpretation. In addition, recommendations made by CDSS regarding medication changes that are not clinically relevant undermine the trustworthiness of the recommendations and discourage clinicians utilising systems (Dalton et al., 2020). Therefore, it is particularly important that trained health care providers evaluate the validity of the predicted results. Furthermore, to facilitate this proposed model, it is important that clinicians understand the patient's goals of care, and how they can contribute to achieving patients' preferred goals through shared-decision making and goal-directed medication reviews. These individualised approaches using the proposed comprehensive intervention bundle provide a promising strategy to achieve precision medicine in polypharmacy by bringing together bioinformatics and clinical judgement to select the medications, doses and formulations most likely to help people with polypharmacy achieve their therapeutic goals. The use of quality indicators will enable healthcare providers to promote further quality improvement activities (Fujita et al., 2018).

## Conclusion

Precision medicine is an approach to maximise the effectiveness of disease treatment and prevention and minimise harm from medications by taking into account relevant demographic, clinical, genomic and environmental factors in making treatment decisions. In people with polypharmacy, the complexity of these factors influencing response to medicines as well as limited direct evidence from human studies make achieving precision medicine challenging. To address this, we proposed a novel approach to involve precision medicine as part of medication review for patients with polypharmacy. For this model to be implemented in routine clinical practice, the integration of the comprehensive intervention bundles into the eMR is necessary, using bioinformatic approaches on a wide range of data to predict the effects of polypharmacy regimens on an individual. In addition, there is a need to train clinicians to interpret the results of the data from sources that include pharmacogenomic testing, DDI prediction and physiological-pharmacokinetic-pharmacodynamic modelling to inform their medication reviews. Future studies are needed to evaluate the efficacy of the model and to test generalisability so that it can be implemented at scale, improving outcomes from polypharmacy.

**Open peer review.** To view the open peer review materials for this article, please visit http://doi.org/10.1017/pcm.2023.10.

**Data availability statement.** All datasets used for the analysis are publicly available at the corresponding references.

**Author contributions.** K.F. and S.N.H. developed the main conceptual ideas for the paper with critical input from N.M., L.K.O. and J.M. All authors contributed to the draft and provided critical revisions. All authors read and approved the final manuscript.

**Financial support.** This research received no specific grant from any funding agency, commercial or not-for-profit sectors. K.F. is supported by NHMRC APP 117 4447. L.K.O. is supported by the Rothwell Fellowship in Geriatric Pharmacotherapy and Penney Ageing Research Unit. N.M. is supported by the Rothwell Fellowship in Geriatric Pharmacotherapy and NHMRC APP 117 4447. J.M. is supported by the Penney Ageing Research Unit.

**Competing interest.** S.N.H. developed and continues to lead an active research program on the Drug Burden Index. The Goal-directed Medication review Electronic Decision Support System (G-MEDSS), which includes a Drug Burden Index calculator, was developed by L.K.O. under the supervision of S.N.H., and is under consideration for commercialisation. The other authors declare none.

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
