## [Reviewer Report]

Professor Dame Anna F Dominiczak,

Editor in Chief, Cambridge Prisms: Precision Medicine

Dear Professor Dominiczak,

We are delighted to submit our invited review article entitled, “Polypharmacy and precision medicine” for consideration for publication in Cambridge Prisms: Precision Medicine.

In this narrative review, we aim to bring together the latest research findings to achieve precision medicine in the context of polypharmacy. Specifically, this review aims to 1) summarise challenges in achieving precision medicine specific to polypharmacy; 2) synthesise the current approaches to precision medicine in polypharmacy; 3) provide a summary of the literature in the field of prediction of unknown drug-drug interaction (DDI); and 4) propose a novel approach to provide precision medicine for patients with polypharmacy. 

Given the growing numbers of patients with multiple conditions who are taking multiple medications, we believe that this review could be of benefit to clinicians and researchers in fields that range from precision medicine and clinical pharmacology to geriatric medicine and primary care.

This manuscript has not been published and is not under consideration for publication elsewhere. All authors agree with the content of the manuscript.

Sincerely,

Sarah Hilmer AM FAHMS

BScMed (Hons) MBBS(Hons) FRACP PhD

Head of Department Clinical Pharmacology and Senior Staff Specialist Aged Care, Royal North Shore Hospital

Conjoint Professor of Geriatric Pharmacology, Northern Clinical School, Faculty of Medicine and Health, University of Sydney

Email: sarah.hilmer@sydney.edu.au

---

## [Reviewer Report]

*Comments to Author*: This is a narrative review examining the various ways that the precepts of precision medicine could be applied to patients with multimorbidity who experience polypharmacy i.e. mostly older people. The aims of the review are clearly laid out and focused. The review is clearly written and sufficiently comprehensive and up-to-date. There is a clearly explained structure as to how precision medicine principles could be applied to an often complex and multifaceted clinical group of patients that incorporates explicit criteria for inappropriate prescribing, DBI calculation, drug-drug interaction data, drug-disease interaction data, pharmacokinetic data, pharmacodynamic data and pharmacogenetic data using well designed and highly functional CDSS.

There are a few minor deficiencies. The term ‘machine learning’ is used quite a lot: it should be clearly defined for the readers not familiar with the term. The sample case summarized in Box 1: several points are raised relating to the list of medications taken by the patient, such as drug-drug interactions, drug-gene interactions etc. It would be useful to go the extra steps and suggest the specific medication optimizing steps that a precision medicine report such as this would make in this particular patient. Most busy prescribers would be rather turned off a report that highlights numerous potential points of drug adversity and would prefer to see a bullet pointed advice report reflecting what an expert in geriatric multimorbidity/polypharmacy would actually recommend in this particular case. Table 1 refers to several mouse studies of the effects of polypharmacy-laden diet versus control diet. Some details of the particular drug combinations making up the ‘polypharmacy’ regimens would be helpful.

---

## [Reviewer Report]

*Comments to Author*: This is an exellent and comprehensive manuscript on the various aspects to manage polypharmacy. 

In addition to the comments of reviewer 1, it would be worth to add a short paragraph chapter 3.2 and 4 on future consideration of drug-drug-gene interactions, in other words how could be both, DDI and DGI be managed as combined parameters affecting drug response.

---

## [Reviewer Report]

Dear Professor Dominiczak,

We would like to thank you, Professor Ingolf Cascorbi (the handling editor) and the reviewer for your insights. We have addressed each of the comments and included the corresponding changes in our revised manuscript. We believe the resubmitted manuscript is stronger and hope that it will be suitable for publication in Cambridge Prisms: Precision Medicine. We have created and uploaded a graphical abstract but this is not currently visible on the proofs. 

I note that the topic and keyword options for this new journal do not currently include 'polypharmacy', 'interactions', 'geriatric' or 'ageing', which would best describe our paper. 

We are happy to address any outstanding issues.

Kind regards,

Sarah Hilmer on behalf of all authors.